# Optimization of Liposomes for Antigen Targeting to Splenic CD169^+^ Macrophages

**DOI:** 10.3390/pharmaceutics12121138

**Published:** 2020-11-25

**Authors:** Maarten K. Nijen Twilhaar, Lucas Czentner, Joanna Grabowska, Alsya J. Affandi, Chun Yin Jerry Lau, Katarzyna Olesek, Hakan Kalay, Cornelus F. van Nostrum, Yvette van Kooyk, Gert Storm, Joke M.M. den Haan

**Affiliations:** 1Department of Molecular Cell Biology and Immunology, Amsterdam University Medical Center, Amsterdam Infection and Immunity Institute, Vrije Universiteit Amsterdam, 1081 HZ Amsterdam, The Netherlands; m.nijentwilhaar@amsterdamumc.nl (M.K.N.T.); j.grabowska@amsterdamumc.nl (J.G.); a.affandi@amsterdamumc.nl (A.J.A.); k.olesek@amsterdamumc.nl (K.O.); h.kalay@amsterdamumc.nl (H.K.); y.vankooyk@amsterdamumc.nl (Y.v.K.); 2Department of Molecular Cell Biology and Immunology, Amsterdam University Medical Center, Cancer Center Amsterdam, Vrije Universiteit Amsterdam, 1081 HZ Amsterdam, The Netherlands; 3Department of Pharmaceutics, Faculty of Science, Utrecht University, Universiteitsweg 99, 3584 CG Utrecht, The Netherlands; l.czentnercolomo@uu.nl (L.C.); c.y.lau@uu.nl (C.Y.J.L.); c.f.vannostrum@uu.nl (C.F.v.N.); g.storm@uu.nl (G.S.); 4Department of Biomaterials, Science and Technology, Faculty of Science and Technology, University of Twente, 7522 NB Enschede, The Netherlands

**Keywords:** cancer vaccination, liposome, targeting, GM3, macrophage, CD169, Siglec-1, sialoadhesin, T cells

## Abstract

Despite promising progress in cancer vaccination, therapeutic effectiveness is often insufficient. Cancer vaccine effectiveness could be enhanced by targeting vaccine antigens to antigen-presenting cells, thereby increasing T-cell activation. CD169-expressing splenic macrophages efficiently capture particulate antigens from the blood and transfer these antigens to dendritic cells for the activation of CD8^+^ T cells. In this study, we incorporated a physiological ligand for CD169, the ganglioside GM3, into liposomes to enhance liposome uptake by CD169^+^ macrophages. We assessed how variation in the amount of GM3, surface-attached PEG and liposomal size affected the binding to, and uptake by, CD169^+^ macrophages in vitro and in vivo. As a proof of concept, we prepared GM3-targeted liposomes containing a long synthetic ovalbumin peptide and tested the capacity of these liposomes to induce CD8^+^ and CD4^+^ T-cell responses compared to control liposomes or soluble peptide. The data indicate that the delivery of liposomes to splenic CD169^+^ macrophages can be optimized by the selection of liposomal constituents and liposomal size. Moreover, optimized GM3-mediated liposomal targeting to CD169^+^ macrophages induces potent immune responses and therefore presents as an interesting delivery strategy for cancer vaccination.

## 1. Introduction

Cancer development is started by the malignant transformation of cells caused by the successive accumulation of mutations that alter protein expression and the behavior of cells [1]. Malignant cells express mutated proteins, proteins that are in physiological circumstances exclusively expressed by other cell types (cancer/testis antigens) or that are normally expressed at very low levels [2,3,4]. The immune system and especially CD8^+^ T cells can recognize peptides derived from mutated or aberrantly expressed proteins that are presented in the context of MHC class I, to subsequently exert an anti-tumor response [3]. The power of the immune system is illustrated by the effectiveness of checkpoint inhibitor immunotherapies in patients with melanoma, which can lead to the significant prolongation of survival and sometimes even to complete tumor regression [5]. However, these beneficial responses are only observed in a minority of patients and can be linked to the presence of a pre-existing immune response [6,7]. To enhance the patients’ response to checkpoint inhibitors, a combination treatment with a cancer vaccine to elicit de novo or to reinvigorate T-cell responses to tumor antigens is proposed to achieve further clinical improvement [8,9,10]. Cancer vaccines should lead to strong antigen presentation by, and the activation of, dendritic cells (DCs), since these are essential for CD8^+^ T-cell priming. However, classical vaccines that consist of protein, peptide, RNA or DNA and are supplemented with an adjuvant do not always result in robust antigen presentation by DCs and thus lead to suboptimal CD8^+^ T-cell responses [9]. In addition, vaccination procedures using ex vivo antigen-loaded and matured monocyte-derived DCs have also proven to be effective in some patients. However, despite large interest and it being quite extensively investigated, this approach remains time-consuming and costly [11,12]. An alternative approach is to design a vaccine that consists of a tumor antigen and adjuvant and additionally has an enhanced capacity to be taken up by DCs [13,14].

Liposomes are versatile delivery vehicles for vaccines since the core can be used to package hydrophilic vaccine components, while the membrane can be used to transfer hydrophobic vaccine components [15]. Moreover, liposomal encapsulation prevents the free distribution of antigen, which should be avoided, as antigens that are administered in a soluble form are taken up by non-professional antigen-presenting cells (APCs) [16,17]. Subsequently, an initial T-cell activation phase can be followed by the induction of tolerance, which is detrimental for the anti-tumor response [18].

Upon liposomal vaccination, phagocytic cells such as DCs take up liposomes at the site of injection and subsequently migrate to secondary lymphoid organs. Alternatively, liposomes drain to the lymph nodes, or the spleen when the liposomes are administered intravenously (IV), after which the liposomes are taken up by tissue-resident APCs [19,20]. While the uptake of non-modified liposomes relies on the intrinsic capacity of APCs to take up particulate antigens, liposomes that contain targeting moieties enable more specific targeting to particular types of APCs and thereby augment liposomal payload delivery and T-cell priming [21,22,23,24].

Multiple cell types within the myeloid compartment are considered as potential targets for cancer vaccines, such as different types of DCs and Langerhans cells (LCs), that can be targeted via their DEC205, Clec9a, DC-SIGN or langerin surface receptors [21,22,23,24]. In addition, recent interest in vaccine targeting was drawn to CD169, also known as Siglec-1 or sialoadhesin, which is the first discovered member of the Siglec family (sialic-acid-binding immunoglobulin-like lectins) [25]. CD169 is highly expressed on metallophilic marginal zone macrophages (MΦ) and subcapsular sinus MΦ, which are strategically located in the spleen and lymph nodes, respectively [26]. These phagocytic cells capture particulate antigens, such as viruses (e.g., human immunodeficiency virus (HIV)) and extracellular vesicles (EVs) [27,28]. Mechanistically, CD169 recognizes α2,3-linked sialic acid-containing gangliosides, such as GM3 [29,30], which are indeed highly enriched in HIV and EVs [31,32,33,34,35,36]. We and others have previously shown that CD169^+^ MΦ pose as an attractive target for cancer vaccines. Upon the targeting of antibody–antigen complexes to CD169, potent cytotoxic T-cell responses were observed. This was dependent on a close collaboration with, and antigen transfer to, conventional DC1 (cDC1), which are the most efficient stimulators of CD8^+^ T-cell responses [37,38]. Moreover, the targeting of liposomes specifically to CD169^+^ MΦ using synthetic ligands resulted in robust iNKT-cell activation and CD8^+^ T-cell activation [39,40].

In the present study, we aimed to specifically target CD169^+^ MΦ using GM3-containing liposomes and to optimize targeting by altering liposomal constituents and liposomal size. Furthermore, we tested the ability of our optimized liposomal vaccines to elicit antigen-specific T-cell responses by the delivery of the immunogenic OVA_247–279_-peptide, which contains the OVA_262–276_ CD4^+^ and OVA_257–264_ CD8^+^ T-cell epitopes. Our data indicate that GM3-containing liposomes that target to CD169^+^ MΦ are a promising antigen delivery vehicle for use in cancer vaccination.

## 2. Materials and Methods

### 2.1. Liposome Preparation and Charactarisation

Liposomes were prepared using the thin lipid film hydration technique followed by extrusion. A mixture of phosphatidylcholine, phosphatidylglycerol (Lipoid GmbH, Ludwigshafen, Germany) and cholesterol (Sigma Aldrich, Darmstadt, Germany) at a molar ratio of 3.8:1:2.5 was dissolved in methanol/chloroform (2:1). Then, 15 µmol of the lipid mixture was transferred to a 50 mL round bottom flask. Where indicated, it was supplemented with GM3 (1, 3 or 5 mol%) (Avanti Polar, Alabaster, AL, US) and/or DSPE-PEG2000 (5 mol%); 0.1 mol% of the lipophilic fluorescent tracer DiD (1′-dioctadecyl-3,3,3′,3′-tetramethyl indodicarbocyanine, Life Technologies, Frederick, MD, USA) was incorporated into all the liposomal preparations. After organic phase evaporation under reduced pressure with a rotary rotavapor, the lipid films were hydrated using 10 mM HEPES buffer at pH 7.4, and where indicated, the buffer contained 3.5 mg of OVA-peptide (OVA_247–279_) or fluorescent 5-FAM-Lys-OVA-peptide (5-FAM-Lys-OVA_247–279_). Subsequently, the liposomes were sized by sequential extrusion through polycarbonate filters (800/600 nm or a 400/200 nm filter combination) using high-pressure nitrogen. The liposomes were concentrated and separated from soluble peptide by ultracentrifugation at 200,000 g (Beckman Coulter). The liposomal pellet was resuspended in 10 mM HEPES buffer at pH 7.4 and washed a second time. After the second centrifugation, the liposomal pellet was resuspended in 1 mL of 10 mM HEPES buffer at pH 7.4.

The phosphate content of the liposomal preparations was determined through an acidic digestion according to Rouser et al. [41]. Liposome sample dilutions and known amounts of a 0.5 mM KH_2_PO_4_ standard solution were transferred into clean glass tubes, and the solvent was completely evaporated using a heat block at 220 °C. Subsequently, 0.3 mL of perchloric acid was added to each tube and placed in the heat block for 60 min or until the yellow color disappeared. When cooled down, 1 mL of water, 0.5 mL of molybdate solution and, subsequently, 0.5 mL of ascorbic acid solution were added and agitated on a vortex. The tubes were placed in a warm water bath for 5 min and then cooled down. Next, the absorbance of the samples and standard was measured at 797 nm, using a spectrophotometric microplate reader (BMG SPECTROstar Nano, De Meern, the Netherlands). The hydrodynamic diameter and polydispersity index (PDI) were measured by dynamic light scattering (DLS) using a Malvern Zetasizer Nano S (Malvern Instruments, Malvern, UK) equipped with a He-Ne 633 nm laser configured with a scattering angle of 173°. The measurements were performed using Malvern’s Zetasizer v7.13 software (Malvern Instruments, Malvern, UK). A minimum of 3 measurements were performed per sample, and the measurement duration was automatically set by the software depending on the samples’ characteristics. A viscosity of 0.8872 centipoises (cP) and refractive index (RI) of 1.330 for the dispersant and an RI of 1.590 and absorption of 0.010 for the material in suspension were set on the software. The samples were diluted to a final phospholipid concentration of 0.15 mM in 10 mM HEPES buffer, pH 7.4, and measured at 25 °C. A PDI < 0.3 was considered acceptable, as it indicated a homogenous size distribution. The zeta-potential was determined using Zetasizer Nano Z (Malvern Instruments, Malvern, UK) and Malvern’s Zetasizer v7.13 software. A minimum of 3 measurements were performed per sample, and the measurement duration was automatically set by the software depending on samples’ characteristics. The samples were diluted to a final phospholipid concentration of 0.15 mM in 0.3 or 1 mM HEPES buffer, pH 7.4, and measured at 25 °C. A viscosity of 0.8872 cP, RI of 1.330 and dielectric constant of 78.5 for the dispersant and an RI of 1.590 and absorption of 0.010 for the material in suspension were set on the software.

### 2.2. Analysis of Peptide Encapsulation

OVA_247–279_ and fluorescent 5-FAM-Lys-OVA_247–279_ were synthesized in house and were confirmed to be of high purity (>90%) using ultra-performance liquid chromatography (UPLC) (Table 1, Appendix A). Control and 3 mol% GM3-containing liposomes were prepared as described above but hydrated in buffer containing 5-FAM-Lys-OVA_247–279_. The liposomes (1.25 mM) were treated with 2% Triton X-100 at 37 °C for 1 h. Subsequently, 5-FAM was excited at 492 nm, and the emission at 517 nm was measured with a Horiba Fluorolog fluorometer FL3-21 (Horiba Jobin Yvon, Longjumeau Cedex, Paris, France). The emission was compared to a standard curve of known peptide concentrations spiked in 1.25 mM liposomes that were treated similarly.

### 2.3. CD169 Fc ELISA

The liposomes (25 µM phosphate) were coated in ethanol on Immuno MaxiSorp plates (NUNC, Roskilde, Denmark) and incubated overnight. The plates were blocked in 1% bovine serum albumin (BSA; Fraction V, Fatty acid free, Calbiochem, San Diego, CA, USA) in phosphate-buffered saline (PBS) and incubated with CD169 Fc or its mutant form (CD169 Fc R97A) (2 μg/mL) for 1 h at room temperature (kindly provided by Prof. Dr. P.R. Crocker, University of Dundee) [41]. Peroxidase-labelled goat anti-human IgG was added for 30 min at room temperature. After a final washing step, 3,3′,5,5′-tetramethylbenzidine (TMB) (Sigma-Aldrich) was added as a substrate, and the optical density (OD) was measured in a microplate absorbance spectrophotometer (Biorad, Hercules, CA, USA) at 450 nm.

### 2.4. Liposome Binding to TSn Cells

The TSn cell line was generated by the lentiviral transduction of the monocytic cell line THP-1 with human Siglec-1 (kindly provided by Dr. A.P. Heikema, Erasmus MC) [42]. Cells were routinely tested for mycoplasma and were maintained in RPMI-1640 (Gibco, Life Technologies, Paisley, UK) supplemented with 10% heat-inactivated fetal calf serum (FCS) (Biowest, Manassas, VA, USA), penicillin, streptomycin and ultraglutamine (Thermo Fisher Scientific, Dreieich, Germany) at 37 °C under 5% CO_2_. Prior to the binding and uptake experiments, 100,000 cells were plated and washed once. Subsequently, liposomes were added to the cells (in various concentrations, based on phospholipid contents, as indicated in the figures). After 45 min of incubation at 4 or 37 °C, unbound liposomes were washed away, and the cells were stained with the Fixable Viability Dye eFluor 780 (eBioscience, San Diego, CA, USA). Following fixation, using 2% paraformaldehyde (PFA) (Electron Microscopy Sciences, Hatfield, PA, USA), the cell-bound liposomal signals in single live cells were quantified by flow cytometry analysis (Aurora, Cytek, Amsterdam, the Netherlands) and analyzed using FlowJo v8 (Tree Star, Ashland, OR, USA).

### 2.5. Enzymatic Spleen Digestion

Spleens from C57BL/6 wild-type (WT) or W2QR97A mutant mice (further information can be found under “animal experiments”) were mechanistically dissociated and subsequently digested using a mixture of 3 mg/mL lidocaine, 2 Wünsch units/mL Liberase TL (Roche, Mannheim, Germany) and 50 mg/mL DNase I (Roche) for 12 min at 37 °C with continuous stirring [38]. Subsequently, cold medium (RPMI-1640 (Gibco, Life Technologies) supplemented with 10% heat-inactivated FCS (Biowest), 10 mM ethylenediaminetetraacetic acid (EDTA), 20 mM HEPES and 50 µM 2-mercaptoethanol) was added, and digestion was continued for an additional 10 min at 4 °C with continuous stirring. After digestion, red blood cells were lysed using ammonium-chloride-potassium lysis buffer, and the splenocytes were filtered through a 70–100 µm filter.

### 2.6. In Vitro Liposome Uptake

Enzymatically digested splenocytes were plated and washed once. Liposomes were diluted in PBS and added to 3 million cells in the concentrations indicated in the figures and incubated for 45 min at 37 °C. The DiD signal in various immune cell populations was assessed after flow cytometry staining as described below.

### 2.7. In Vivo Liposome Uptake

Mice were IV injected with liposomes (as indicated in the figures) supplemented with 25 µg of Polyinosinic:polycytidylic acid (poly (I:C)) (low molecular weight (LMW), InvivoGen, Toulouse, France) and 25 µg of anti-CD40 antibody (clone 1C10). After 2 h, the mice were sacrificed, and the enzymatically digested splenocytes were used for flow cytometry analysis as described below.

### 2.8. Fluorescent Microscopy

Mice were IV injected with liposomes, and spleens were harvested after 2 h. Subsequently, the splenic tissue was cryopreserved in liquid nitrogen and cut into slices of 5–6 µm thickness using a CryoStar NX70 (Thermo Fischer Scientific). The tissue sections were blocked using 10% normal goat serum in PBS and stained with anti-CD169 (clone SER-4, in-house produced), anti-B220 (clone RA3-6B2, BD Biosciences, San Jose, CA, USA) and DAPI. The sections were analyzed using a fluorescent microscope (Zeiss, Oberkochen, Germany) and processed using the Fiji software v1.51. 

### 2.9. In Vivo T-Cell Priming

Mice were IV or subcutaneously (SC) injected with liposomes and supplemented with 25 µg of poly(I:C) (low molecular weight (LMW), InvivoGen) and 25 µg of anti-CD40 antibody (clone 1C10). On Day 7, the mice were sacrificed and the spleens were isolated. Where indicated, blood was collected via a cardiac puncture. The spleens were mashed through a 70–100 µm filter to obtain a single-cell suspension. Red blood cells were lysed using ammonium-chloride-potassium lysis buffer. Subsequently, the cells were used for peptide restimulation and flow cytometry analysis.

### 2.10. Flow Cytometry and Antibodies

Splenocytes were incubated with 10 µg/mL of anti-CD16/32 (clone 2.4G2, in-house produced) for 15 min at 4 °C to block unspecific Fc receptor binding and stained for 30 min using the following antibodies: anti-CD169 (clone SER-4, in-house produced), anti-B220 (clone RA3-6B2, BD Biosciences), anti-F4/80 (clone T45-2342, BD Biosciences), anti-CD8a (clone 53-6.7, BD Biosciences), anti-CD11c (clone HL3, BD Biosciences), anti-I-A/I-E (clone M5/114.15.2, BD Biosciences) and the Fixable Viability Dye eFluor 780 (eBioscience, San Diego, CA, USA) diluted in PBS/0.5% bovine serum albumin (BSA).

Alternatively, splenocytes were stained with anti-XCR1 (clone ZET, BioLegend, San Diego, CA, USA), anti-CD11c (clone HL3, BD Biosciences), anti-I-A/I-E (clone M5/114.15.2, BioLegend), anti-Ly6G (clone 1A8, BioLegend), anti-BST2 (clone 129C1, BioLegend), anti-CD11b (clone M1/70, BioLegend), anti-CD169 (clone SER-4, in-house produced), anti-Siglec-H (clone 551, BioLegend), anti-F4/80 (clone T45-2342, BD Biosciences), anti-CD8a (clone 53-6.7, BD Biosciences), anti-Sirpα (clone P84, BioLegend); the lineage markers anti-CD3e (clone 145-2C11, BioLegend), anti-CD19 (clone 6D5, BioLegend) and anti-NK1.1 (clone PK136, BioLegend); and the Fixable Viability Dye eFluor 780 (eBioscience) diluted in PBS/0.5% BSA.

OVA-specific CD8^+^ T cells were identified by staining with anti-CD8a antibody (clone 53-5.6, BD Biosciences), anti-CD44 antibody (clone KM81, ImmunoTools, Friesoythe, Germany) and PE-labelled H-2K^b^/SIINFEKL tetramer (LUMC, Leiden, The Netherlands) at 37 °C for 60 min.

Splenocytes were restimulated for 5 h with OVA_257–264_ in the presence of Golgiplug (BD Bioscience) or 20 h with OVA_262-276_, followed by 5 h of incubation with Golgiplug. Intracellular IFNγ was detected by staining with an anti-CD11a antibody (clone M17/4, eBioscience) and anti-CD8 antibody (clone 53–5.6, BD Biosciences), or anti-CD4 antibody (clone GK1.5, eBioscience). After surface staining, the cells were fixed using 2% PFA (Electron Microscopy Sciences, Hatfield, PA, USA). For intracellular IFNγ detection, the cells were permeabilized using 0.5% saponin buffer and stained with anti-IFNγ antibody (clone XMG1.2, eBioscience) for 30 min at 4 °C.

### 2.11. Animal Experiments

C57BL/6 WT mice were bought from Charles River or bred in house. W2QR97A-mutant animals, also referred to in the text as CD169-mutant animals, harboring two amino acid substitutions (Trp2 to Gln, and Arg97 to Ala) in the CD169 receptor, were kindly provided by Prof. Dr. P.R. Crocker, University of Dundee, Scotland, and bred in house [41]. Male and female mice were used for the liposomal uptake studies. For the T-cell priming studies, female mice 8–12 weeks of age were used. All the animal experiments were approved by the National Committee for Animal Experiments (CCD AVD1140020171024, 10 April 2017) and the local animal welfare body, Vrije Universiteit, Amsterdam UMC.

### 2.12. Statistical Analysis

Statistical analysis was performed using GraphPad Prism v8 (GraphPad, San Diego, CA, USA). Statistical differences were determined using a one-way ANOVA with a Tukey post-hoc test, a Dunnett post-hoc test when a statistical difference from a control formulation was determined, or a Bonferroni post-hoc test when a statistical difference between preselected groups was found. A p value of 0.05 or lower was considered significant.

## 3. Results

### 3.1. Influence of GM3 Incorporation

In order to determine the optimal mol% of GM3 for binding to CD169-expressing cells, we prepared 150–200 nm liposomes containing 0–5 mol% GM3. The liposomes exhibited similar sizes and negative charges (Table 2).

To confirm that GM3-containing liposomes could bind to mouse CD169, we analyzed binding to a recombinant mouse CD169 Fc in an ELISA-based assay. The specificity of the interaction was warranted by assessing binding to a mutant form of recombinant mouse CD169 Fc (R97A) with an impaired capacity to bind to α2,3-linked sialic acids [41]. The incorporation of 0–5 mol% GM3 in our liposomal preparations resulted in a dose-dependent enhanced ability of binding to CD169 Fc, while binding to its mutant form was not detected (Figure 1A).

Next, we investigated liposome binding to THP-1 cells that stably express high levels of human CD169 (CD169 is also known as sialoadhesin (Sn), and these cells are further referred to as (TSn)) [42]. Liposomes were incubated with cells at 4 °C (Figure 1C) or 37 °C (Figure 1B,D). The incorporation of either 3- or 5 mol% GM3 resulted in superior binding, compared to the binding of control liposomes, in all the tested concentrations (illustrated in Figure 1B and quantified in Figure 1C,D). By contrast, liposomes containing 1 mol% GM3 only significantly improved binding at the highest concentration tested, compared to the control liposomes.

We subsequently tested the in vitro binding of GM3-containing liposomes to mouse splenic CD169^+^ MΦ, red pulp macrophages (red pulp MΦ), dendritic cells types 1 and 2 (cDC1 and cDC2, respectively) and B cells (gating strategy in Appendix A). In addition, we assessed the specificity of CD169 targeting by liposome incubation with splenocytes isolated from mice expressing a mutant form of CD169 containing two mutations (W2QR97A) that impair the binding to α2,3-linked sialic acids [41]. GM3-containing liposomes strongly bound to CD169^+^ MΦ, and this binding was not observed with mutant CD169^+^ MΦ or other cell types (Figure 1E). The liposomal incorporation of 1 mol% GM3 was found to enhance binding compared to a control formulation, while the incorporation of 3–5 mol% GM3 resulted in superior targeting (Figure 1E).

Together, these results indicate that in vitro, 3–5 mol% GM3-containing liposomes exhibit superior binding properties to both mouse and human CD169-expressing cells, compared to control liposomes that lack GM3.

To further corroborate our in vitro results with in vivo evidence, we injected mice IV with GM3-containing liposomes, supplemented with adjuvant. Then, 2 h after vaccination, murine spleens were isolated and analyzed using flow cytometry and fluorescent microscopy. In order to assess which splenic APCs took up liposomes in vivo, we gated on live, lineage^−^, Ly6G^−^ and MHCII^+^/autofluorescent^+^ (AF) cells (Appendix A). The gated cells from the individual mice were pooled, and we applied unsupervised high-dimensionality reduction clustering analysis using *t*-distributed stochastic neighbor embedding (tSNE) to cluster splenic APC populations based on their cell-surface marker expression and to visualize the capacity of these APC populations to take up liposomes in vivo.

Interestingly, all GM3-containing liposomes almost exclusively co-localized with a single population, while control liposomes were also taken up by a second population (Figure 2A as indicated by the arrow or dashed line, respectively). As expected, APC clusters from non-injected mice did not show a fluorescent DiD signal (Figure 2A). Subsequently, we manually gated CD169^+^ MΦ, red pulp MΦ, cDC1, cDC2 and plasmacytoid DC (pDC) (Appendix A) and identified these populations in the tSNE plot to elucidate which cells were responsible for liposome uptake in vivo (Figure 2B). In addition, we quantified the DiD signal in these populations (Figure 2C). As expected, CD169^+^ MΦ were the dominant population responsible for the uptake of GM3-containing liposomes. Although CD169^+^ MΦ also took up control liposomes, the liposomal incorporation of 3–5 mol% GM3 led to an increased uptake. In addition, we detected liposomal binding by red pulp MΦ, albeit to a much lower extent, which was not associated with the GM3 content. In fact, the liposomal incorporation of GM3 tended to diminish uptake by red pulp MΦ. Liposome uptake was not observed for the different DC populations (Figure 2A–C). Microscopic analysis of the splenic tissue sections further corroborated the pivotal role of CD169^+^ MΦ in liposome uptake, as IV-injected liposomes, regardless of their GM3 contents, were found in close proximity to CD169^+^ MΦ (Appendix A).

Thus, IV-injected liposomes are predominantly taken up by CD169^+^ MΦ, and liposomal uptake by these cells can be optimized by the inclusion of 3–5 mol% GM3.

### 3.2. Influence of Liposomal PEG

IV-administered liposomes tend to be rapidly eliminated from the blood circulation, and the liposomal incorporation of DSPE-PEG2000 (further referred to as PEG) is used to slow down the clearance and to enhance the plasma area under the curve [15]. We postulated that liposomal surface-attached PEG might lead to a prolonged exposure of splenic CD169^+^ MΦ to liposomal GM3, resulting in an enhanced targeting efficiency. However, PEG chains on the liposomal surface can also potentially interfere with the interaction between GM3 and CD169 and thereby reduce targeting efficiency. In order to study the effect of PEG on targeting efficiency, we prepared liposomes with 0 or 3 mol% GM3, with or without 5 mol% PEG (Table 3). The inclusion of PEG did not affect the mean liposomal size to a large extent. However, the inclusion of PEG reduced the zeta potential by about 2-fold.

We tested the binding of 0 mol% (control) and 3 mol% GM3-containing liposomes with or without the additional co-incorporation of 5 mol% PEG to WT and mutant CD169 Fc. As observed earlier, the binding of GM3-containing liposomes to WT CD169 Fc was detected. The additional incorporation of PEG completely abolished the interaction between GM3 and CD169 Fc. Control and PEG-containing liposomes did not bind CD169 Fc, and none of the formulations bound to the mutant form of CD169 Fc (Figure 3A).

Subsequently, we assessed liposome binding to TSn, after incubation at 4 °C (Figure 3C) or 37 °C (Figure 3B,D). While GM3-containing liposomes strongly bound to TSn, the additional incorporation of PEG abolished the effect of GM3. The incorporation of PEG in the absence of GM3 did not affect the basal signal compared to a control formulation (illustrated in Figure 3B and quantified in Figure 3C,D). An additional analysis of liposome binding to splenocytes in vitro revealed that GM3-containing liposomes strongly bound to CD169^+^ MΦ compared to control liposomes. However, the co-incorporation of PEG completely abolished the binding of GM3-containing liposomes to CD169^+^ MΦ in vitro (Figure 3E). Next, we analyzed the uptake of liposomes in vivo, 2 h after IV injection. The liposomal co-incorporation of PEG resulted in a significant decrease in the uptake of GM3-containing liposomes by CD169^+^ MΦ, while the low uptake by red pulp MΦ was not affected by liposomal PEGylation (Figure 3F).

Thus, our in vitro results clearly demonstrate that the liposomal inclusion of 5 mol% PEG completely abolishes the interaction between GM3 and CD169, while in vivo, the uptake of GM3-containing liposomes by CD169^+^ MΦ is suboptimal upon the inclusion of 5 mol% PEG in these liposomes.

### 3.3. Influence of Liposomal Size

An increased liposomal size can increase the liposomal payload delivery and thereby its effect, but may also alter liposomal targeting efficiency [43]. In order to study the effect of liposomal size on targeting efficiency, we prepared 0 mol% (control) and 3 mol% GM3-containing liposomes that differed in size (referred to as small and large, respectively) (Table 4).

We tested the binding of small and large GM3-containing liposomes to recombinant mouse CD169 Fc in an ELISA-based assay. As expected, we observed a similar capacity of the small and large GM3-containing liposomes to bind to WT CD169 Fc but not its mutant form (Figure 4A).

Subsequently, we tested liposome binding to human CD169-expressing TSn after incubation at 4 °C (Figure 4C) or 37 °C (Figure 4B,D). We observed that the liposomal incorporation of GM3 in both small and large liposomes resulted in similar binding to TSn. Although significant differences were found for binding between small and large GM3-containing liposomes to TSn, this difference was minor. Both small and large GM3-containing liposomes bound very efficiently to TSn (illustrated in Figure 4B and quantified in Figure 4C,D).

The in vitro binding of liposomes to CD169^+^ MΦ and other splenic cell populations was found to be similar for both small and large GM3-containing liposomes, while no binding for small and large control formulations was found (Figure 4E).

Next, we injected small and large liposomes IV and determined the extent to which the liposomes were taken up by splenic CD169^+^ MΦ after 2 h. Small GM3-containing liposomes were found to be associated with CD169^+^ MΦ to a greater extent than large GM3-containing liposomes. While the incorporation of GM3 resulted in enhanced targeting efficiency for small liposomes, this was not observed for large liposomes. As observed in previous experiments, we also detected some uptake of liposomes by red pulp MΦ. Interestingly, for both small and large liposomes, liposomal GM3 incorporation tended to result in a diminished uptake by red pulp MΦ (Figure 4F).

In conclusion, these experiments show that a two-fold increase in liposomal size results in suboptimal targeting, as it significantly decreases the binding of GM3-containing liposomes to CD169^+^ MΦ in vivo, but not to CD169-expressing cells in vitro.

### 3.4. Immunogenicity of Liposomes Containing Antigen

Since our experiments demonstrated strong binding of small 3 mol% GM3-containing liposomes to CD169-expressing cells in vitro and in vivo, we prepared control and GM3-liposomes that contained an ovalbumin peptide as a model antigen (OVA_247–279_). This long OVA_247–279_ peptide is immunogenic and contains a CD4^+^ and a CD8^+^ T-cell epitope, OVA_262–276_ and OVA_257–264_, respectively. Next, the liposome size, PDI and zeta-potential were determined (Table 5). In addition, we quantified the encapsulation efficiency and loading capacity of the liposomes using fluorescent emission, which was compared to known concentrations of a fluorescent peptide (Table 5 and Appendix A).

In order to confirm that peptide inclusion did not interfere with the binding of control and GM3-containing liposomes to CD169 Fc, we performed an ELISA-based binding assay and found the binding of GM3-containing liposomes, but not control liposomes, to CD169 Fc (data not shown).

Next, we immunized C57BL/6 WT mice with different doses of 0 and 3 mol% GM3-containing liposomes that contained the OVA_247–279_ peptide in the presence of adjuvant. On Day 7 after IV vaccination, the spleens were harvested to determine the frequency of OVA-specific CD8^+^ T cells, which was defined as the percentage of CD8^+^ T cells that bound the H-2K^b^/SIINFEKL tetramer. In addition, the frequency of IFNγ-producing antigen-specific CD4^+^ or CD8^+^ T cells was analyzed after OVA_262–276_ or OVA_257–264_ peptide restimulation, respectively. We observed that OVA-specific CD8^+^ T-cell responses were induced by both control and GM3-containing liposomes in a dose-dependent manner (Figure 5A,B). The liposomal inclusion of GM3 resulted in significantly enhanced CD8^+^ T-cell responses when compared to control liposomes at the highest dose. At the two lower doses, GM3-containing liposomes also resulted in slightly higher CD8^+^ T-cell responses (Figure 5A,B). While immunization with liposomes also induced CD4^+^ T-cell responses, the inclusion of GM3 did not influence the magnitude of the response (Figure 5C).

This initial IV immunization experiment showed the robust induction of CD8^+^ and CD4^+^ T-cell responses with the high dose (22.5 nmol) of GM3-containing liposomes. Since both splenic and lymph node CD169^+^ MΦ are known to capture liposomes, we aimed to elucidate whether immunization via the SC route could elicit an equally robust immune response. Thus, we IV and SC immunized mice with control and GM3-containing liposomes and determined the amount of antigen-specific T cells on Day 7 after immunization. We observed that for the GM3- and control liposomes, IV immunization resulted in superior CD8^+^ and CD4^+^ T-cell responses when compared to SC immunization (Figure 5D–F). In addition to the splenic compartment, strong responses were also observed in the blood after IV immunization (Appendix A).

Our analysis of the peptide incorporation revealed that this dose corresponded to approximately 50 ng of OVA-peptide (Table 5). To benchmark the potency of our optimal liposomal vaccination strategy, we performed an immunization experiment in which we immunized mice with 22.5 nmol of GM3-containing liposomes or 50 ng or 50 μg of soluble peptide. Antigen-specific immune responses were detected in mice immunized with GM3-containing liposomes or the high dose of soluble peptide (Figure 5G–I). Conversely, immunization with the low dose of soluble peptide, which was equal to the amount of peptide in the GM3-containing liposomes (i.e., 50 ng), did not induce noticeable immune activation (Figure 5G–I). Furthermore, immunization with GM3-containing liposomes that contained 1000-fold less peptide (i.e., 50 ng) compared to the high dose of soluble peptide resulted in similar CD4^+^ T-cell responses and only a 2-fold decrease in the activation of CD8^+^ T cells (Figure 5G–I).

Collectively, these data indicate that the inclusion of the targeting moiety GM3 in liposomes improves CD8^+^ T-cell activation compared to a control liposome. Moreover, IV immunization with our optimized liposomes dramatically increases vaccine potency as compared to SC immunization or IV immunization with soluble administered non-targeted peptide.

## 4. Discussion

Cancer (neo) antigens and adjuvants are important components of cancer vaccines that have to be carefully identified and selected [2,3]. In addition, vaccine characteristics that can enhance specific uptake by antigen-presenting cells can be considered as a third factor that determines the efficacy of T-cell priming [44,45]. Our aim was to develop a targeting strategy to increase liposomal cargo delivery to splenic CD169^+^ MΦ and thereby to enhance the potency of T-cell responses. Here, we show that GM3-containing liposomes specifically bind to mouse and human CD169-expressing cells and can be used to deliver antigens to these APCs, to induce antigen-specific T-cell responses. We have evaluated the effects of the amount of incorporated GM3, the presence of surface-exposed PEG and particle size on liposomal targeting and conclude that small (i.e., 200 nm) liposomes with 3 mol% GM3 and without PEG are the most optimal for targeting splenic CD169^+^ MΦ.

Splenic metallophilic marginal zone and lymph node subcapsular sinus MΦ express high levels of CD169 and are known for their capacity to capture particles such as viruses and EVs, which are enriched in gangliosides, including GM3 [31,32,33,34,35,36]. In addition, a new subset of human DCs, characterized by the expression of Axl and Siglec-6 (AS DC), was also shown to express CD169 [46]. We recently demonstrated that ganglioside-containing liposomes can be used to selectively target and activate these human AS DCs in a CD169-dependent manner [47]. In this study, we have optimized targeting to human and murine cells and show that the liposomal incorporation of either 3 or 5 mol% GM3 led to strong and specific binding to CD169-expressing cells in vitro and in vivo. Our results are in line with a study on artificial viral nanoparticles, which showed the binding of GM3-containing nanoparticles to CD169-expressing cells and sequestration in lymph nodes [31,32,48]. In this study, we describe that GM3-containing liposomes can be used to efficiently and specifically target splenic CD169^+^ MΦ in vivo after IV administration and result in the activation of potent T-cell responses. In addition, our results also show the efficient uptake of control liposomes by these CD169^+^ MΦ in vivo, but not in vitro. This supports the notion that CD169^+^ MΦ play a pivotal role as gatekeepers in the spleen to sequester particulate antigens, including passively and actively targeted liposomes, but also demonstrates that in vitro studies do not fully recapitulate this process [27]. The uptake of our anionic control liposomes in vivo could potentially be mediated by complement proteins or factors involved in the coagulation cascade that are inactivated in in vitro systems [49].

In addition to metallophilic marginal zone CD169^+^ MΦ, red pulp MΦ express low levels of CD169 (Appendix A) and also took up control and GM3-containing liposomes, but to a much lower extent. However, liposomal GM3 inclusion was not correlated with enhanced liposome uptake by these red pulp MΦ and even appeared to suppress uptake. Since the inclusion of GM3 results in the very efficient filtering of the liposomes by CD169^+^ MΦ in the marginal zone, fewer liposomes are likely to be available for binding to macrophages in the red pulp. In line with these observations, a study by Saunderson and co-workers demonstrated that CD169^+^ MΦ very efficiently captured EVs, but when CD169^−/−^ mice were used, the EV capture in the marginal zone was decreased, and it increased in the red pulp [28]. This indicates that the CD169^+^ MΦ in the marginal zone are the primary “first pass” cell type to capture liposomes and that therefore less liposomes become available for uptake by red pulp MΦ.

Red pulp MΦ are known to be involved in the blood clearance of larger particles that cannot pass the endothelial slits in the venous sinuses, including damaged red blood cells [26,50,51]. We studied to what extent small and large control and GM3-containing liposomes were taken up by CD169^+^ MΦ and red pulp MΦ. We found that a 2-fold increase in liposomal mean size resulted in a reduced uptake of GM3-containing liposomes by CD169^+^ MΦ. However, a 2-fold increase in liposomal mean size did not correlate with an altered interaction between liposomes and red pulp MΦ, which may have been expected, as fewer liposomes were captured by CD169^+^ MΦ. The explanation for the observed reduced efficiency of targeting to CD169^+^ MΦ remains unclear. Potentially, the increased liposomal size facilitates enhanced protein opsonization and complement activation in the blood, processes that affect the interaction of GM3 with CD169 and can potentially mediate the destruction of liposomal particles [49,52,53,54].

We also studied the effect of surface-exposed PEG on the targeting efficiency of GM3-containing liposomes. The incorporation of 5 mol% PEG completely abolished the binding of liposomes to CD169^+^ MΦ in vitro and significantly diminished the uptake of GM3-containing liposomes in vivo. Liposomal PEGylation is known to increase the circulation time of the liposomes in the bloodstream [55,56]. However, despite a possible increased circulation time, the incorporation of PEG in GM3-containing liposomes did not result in the accumulation of the PEGylated liposomes in splenic CD169^+^ MΦ due to the interference of PEG in the interaction between GM3 and CD169 [44]. These results clearly discourage the inclusion of PEG in liposomal vaccines that target CD169^+^ MΦ and/or DCs via ligand–receptor binding.

While CD8^+^ T cells are major players involved in the anti-tumor response, the additional induction of CD4^+^ T cells has proven to be beneficial for clinical responses [57]. Therefore, cancer vaccines should contain both CD4^+^ and CD8^+^ T-cell epitopes to allow for antigen presentation in MHC class I and II [3]. We determined the potency of vaccination with control and GM3-containing liposomes that contained synthetic OVA-peptide with both epitopes. When mice were vaccinated with liposomes in the presence of a strong adjuvant, we observed a dose-dependent activation of CD8^+^ T-cell responses for GM3-containing liposomes and to a lower extent also for control liposomes. Since the control liposomes were already efficiently taken up by CD169^+^ MΦ in vivo, T-cell priming upon injection with control liposomes was anticipated to occur. Importantly, the liposomal inclusion of GM3 enhanced the degree of both the uptake and subsequent CD8^+^ T-cell priming. These results are in line with our previous studies in which we discovered a collaboration between CD169^+^ MΦ and cross-presenting cDC1 that are specialized in CD8^+^ T-cell priming [38,58]. In these previous studies, we targeted antigens to CD169^+^ MΦ using antibodies and detected antigen transfer from CD169^+^ MΦ to cDC1. Indeed, cDC1 were found to be responsible for CD8^+^ T-cell induction. In this study, we show that immunization with liposomes harboring a natural ligand of CD169 also results in efficient CD8^+^ T-cell priming. Recently, the incorporation of a synthetic CD169 ligand in liposomes was also shown to result in CD8^+^ T-cell priming [40]. Further studies will address the role of cDC1 in the process of CD8^+^ T-cell priming after liposomal targeting to CD169^+^ MΦ.

Finally, our data show that the encapsulation of antigenic peptides in liposomes dramatically increases their immunogenicity when administered IV, while this was not observed when the liposomes were administered SC. Anionic liposomes have been described to drain better to lymph nodes than cationic liposomes, and smaller liposomes (<100 nm), better than larger liposomes (900 nm) [19]. Potentially, the size of our anionic liposomes (230 nm) already hampers efficient lymphatic drainage and thereby leads to suboptimal immune responses. Our studies indicate that the scavenging of anionic liposomes by splenic CD169^+^ MΦ is very efficient after IV administration and that this results in very strong immunogenicity when compared to free peptide. This is in line with other studies in which cationic liposomes and self-assembling nanoparticles exhibited stronger immunogenicity than free antigen [59,60]. Furthermore, the use of nanoparticles allows for the additional incorporation of adjuvant that activates APCs. Interestingly, it has been shown that cellular stimulation with toll-like receptor (TLR) agonists can also enhance cross-presentation by DCs [61,62]. Recent studies have included several different TLRs and stimulator of interferon genes (STING) agonists in liposomes and showed their potential for T-cell activation and tumor protection [59,63]. In future studies, we will investigate the optimal TLR/STING ligands to be incorporated, as adjuvants, in our GM3-containing liposomes.

In conclusion, GM3-containing liposomes bind specifically to human and mouse CD169-expressing cells, and GM3-containing liposomes that contain antigenic peptides stimulate potent immune responses in vivo. Therefore, GM3-containing liposomes are interesting antigen-delivery vehicles to be used in cancer vaccination.

## Figures and Tables

**Figure 1 pharmaceutics-12-01138-f001:**
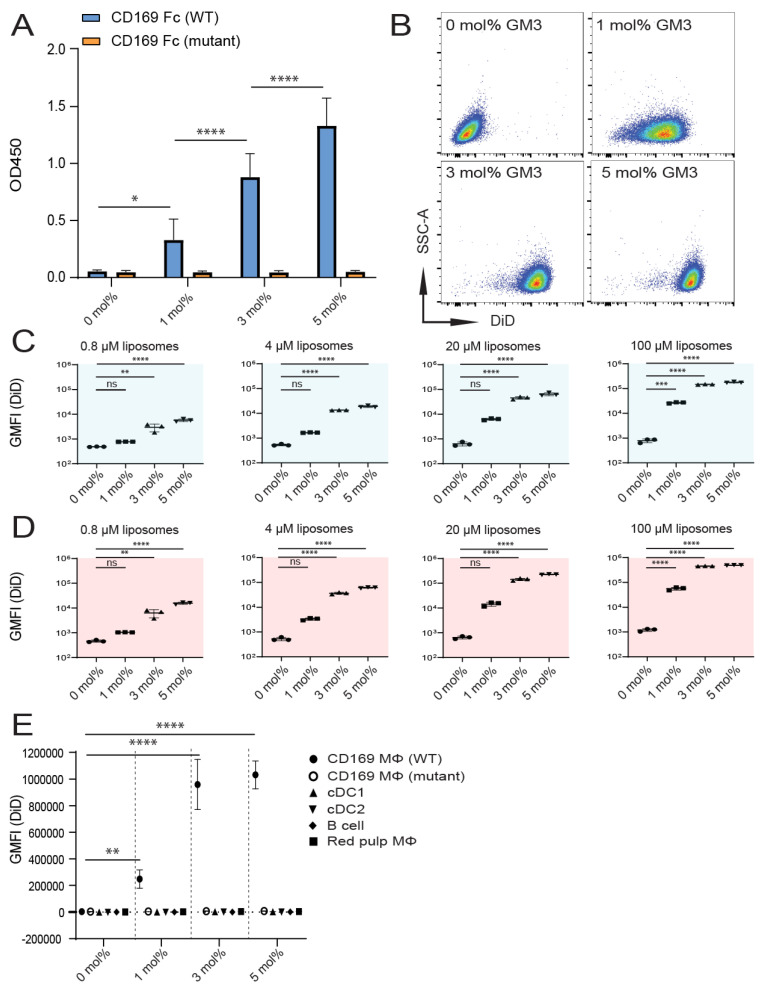
Inclusion of GM3 in liposomes results in specific binding to CD169-expressing cells in vitro. (**A**) Liposomes were coated overnight on an ELISA plate, and binding to wild-type (WT) or mutant mouse CD169 Fc was quantified. Indicated is the average OD450 of 3 independent experiments. (**B**–**D**) 1′-dioctadecyl-3,3,3′,3′-tetramethyl indodicarbocyanine (DiD)-containing liposomes were incubated with TSn at 4 or 37 °C for 45 min. Representative dot plots following incubation on 37 °C (20 µM liposomes) (**B**). Indicated is the average geometric mean fluorescence intensity GMFI ± SD of a technical triplicate after incubation at 4 °C (**C**) and at 37 °C (**D**) (pattern representative of 4 independent experiments). (**E**) DiD-containing liposomes were incubated for 45 min at 37 °C with freshly isolated splenocytes from C57BL/6 WT or W2QR97A-mutant CD169 mice. Subsequently, DiD fluorescence was quantified for distinct cell populations using flow cytometry. Indicated is the GMFI ± SD (*n* = 5) (representative of 2 independent experiments). * *p* < 0.05, ** *p* <0.01, *** *p* < 0.005 and **** *p* < 0.0001, ns: no significance.

**Figure 2 pharmaceutics-12-01138-f002:**
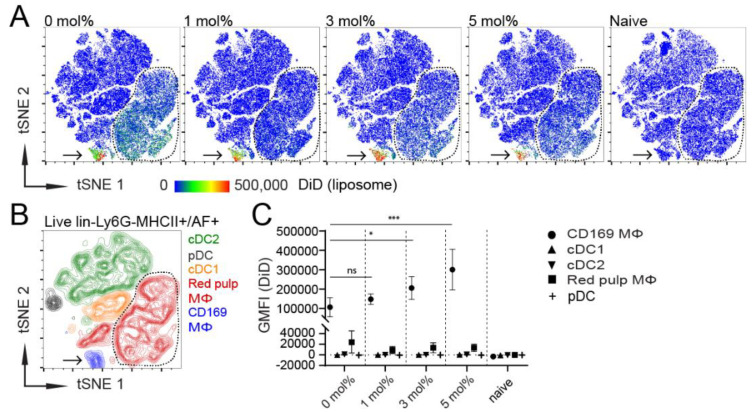
CD169^+^ MΦ are the main antigen-presenting cells (APCs) that take up liposomes, and the inclusion of 3–5 mol% GM3 augments targeting efficiency. (**A**) DiD-containing liposomes (22.5 nmol of phospholipid), supplemented with adjuvant, were intravenously (IV) injected into mice. After 2 h, splenocytes were stained and major splenic APC populations were clustered using high-dimensional data reduction analysis (pre-gated on live, lineage^−^, Ly6G^−^ and MHCII^+^/AF^+^ cells). Liposomal DiD fluorescence in APC populations is depicted for 0, 1, 3 or 5 mol% GM3-containing liposomes and non-injected naive mice. (**B**) CD169^+^ MΦ, red pulp MΦ, cDC1, cDC2 and pDC were manually gated and overlaid to identify the clusters that were obtained by high-dimensional data reduction. Arrow points to CD169^+^ MΦ, and red pulp MΦ are bordered by a line. (**C**) Quantification of DiD fluorescence within various splenic APC populations. Indicated is the average GMFI ± SD (*n* = 6 for CD169^+^ MΦ, red pulp MΦ, cDC1 and cDC2 and *n* = 3 for pDC). * *p* < 0.05, *** *p* < 0.001, ns: no significance.

**Figure 3 pharmaceutics-12-01138-f003:**
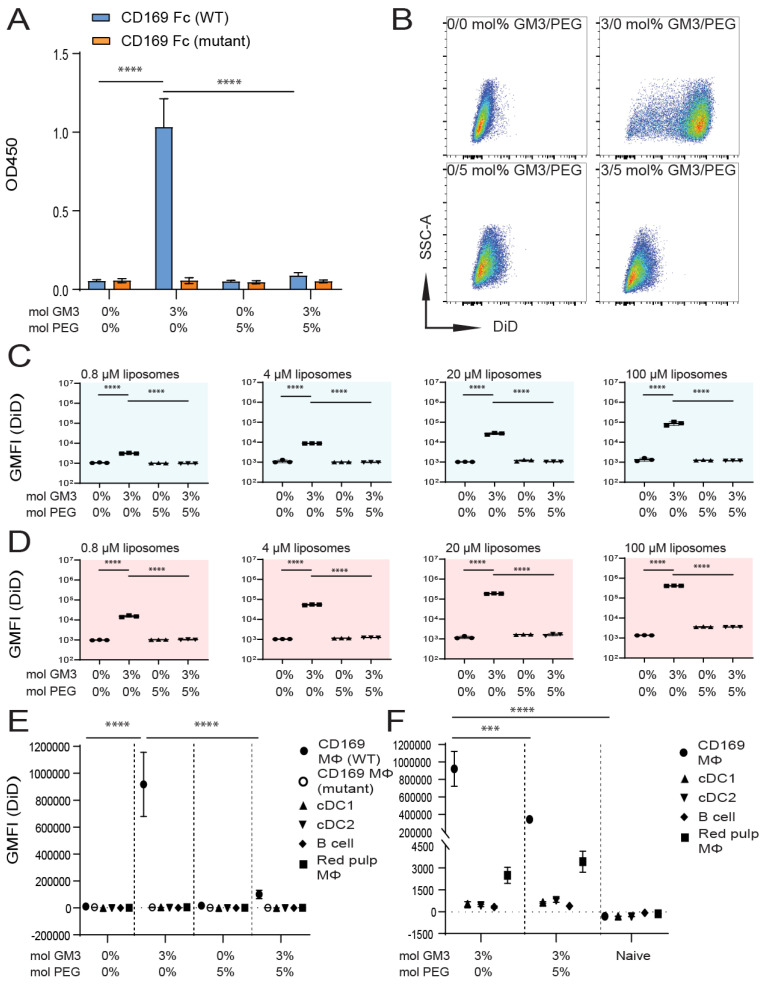
Liposomal incorporation of PEG abolishes the interaction between GM3 and CD169. (**A**) Liposomes were coated overnight on an ELISA plate, and binding to WT or mutant CD169 Fc was quantified. Indicated is the average OD450 of 3 independent experiments. (**B**–**D**) DiD-containing liposomes were incubated with TSn at 4 or 37 °C for 45 min. Representative dot plots following incubation at 37 °C (20 µM liposomes) (**B**). Indicated is the average GMFI ± SD of technical triplicates after incubation at 4 °C (C) and at 37 °C (**D**) (pattern representative of 4 independent experiments). (**E**) DiD-containing liposomes were incubated for 45 min at 37 °C with freshly isolated splenocytes from C57BL/6 WT or W2QR97A-mutant CD169 mice. Subsequently, DiD fluorescence was quantified for distinct cell populations using flow cytometry. Indicated is the GMFI ± SD (*n* = 5) (representative of 2 independent experiments). (**F**) DiD-containing liposomes (90 nmol of phospholipid), supplemented with adjuvant, were injected IV in mice, and after 2 h, splenic cell populations were analyzed for DiD fluorescence. Indicated is the average GMFI ± SD (*n* = 4). *** *p* < 0.005 and **** *p* < 0.0001, ns: no significance.

**Figure 4 pharmaceutics-12-01138-f004:**
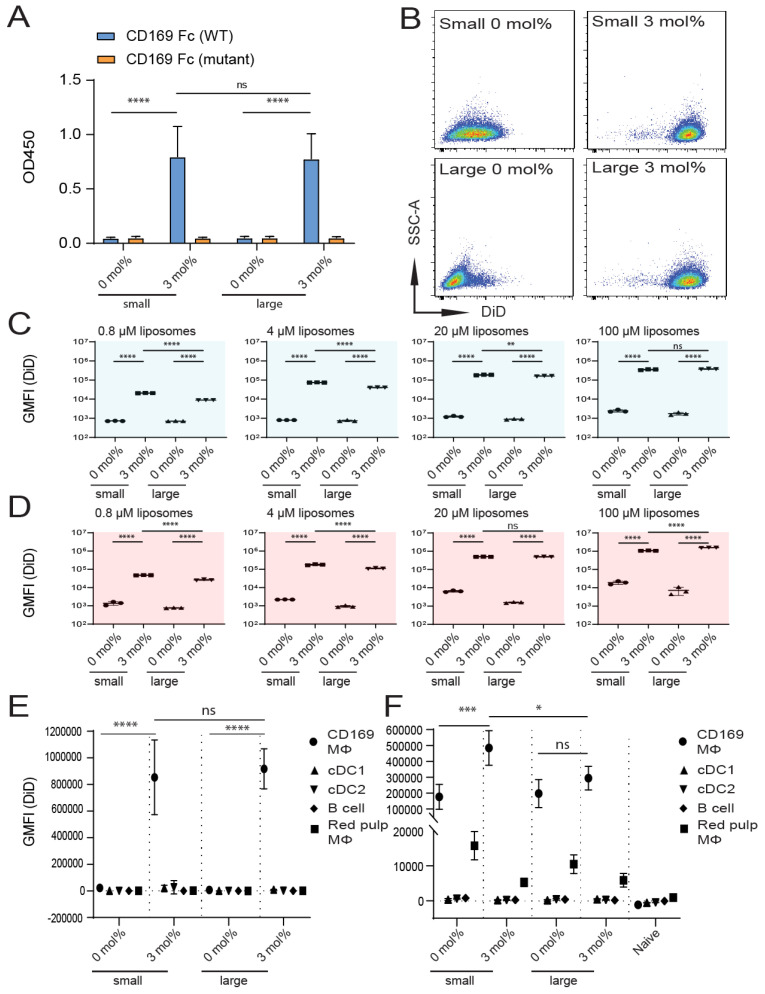
Size of GM3-containing liposomes influences binding to CD169-expressing cells in vivo, but not in vitro. (**A**) Liposomes were coated overnight on an ELISA plate, and binding to WT or mutant CD169 Fc was quantified. Indicated is the average OD450 of 2 independent technical triplicates. (**B**–**D**) DiD-containing liposomes were incubated with TSn at 4 or 37 °C for 45 min. Representative dot plots following incubation at 37 °C (20 µM liposomes) (**B**). Indicated is the average GMFI ± SD of a technical triplicate after incubation at 4 °C (**C**) and at 37 °C (**D**) (representative of 4 independent experiments). (**E**) DiD-containing liposomes were incubated for 45 min at 37 °C with freshly isolated splenocytes from C57BL/6 WT mice. Subsequently, DiD fluorescence was quantified for distinct cell populations using flow cytometry. Indicated is the GMFI ± SD (*n* = 4) (representative of 2 independent experiments). (**F**) DiD-containing liposomes (22.5 nmol of phospholipid) were injected IV in mice, and after 2 h, splenic cell populations were analyzed for DiD fluorescence. Indicated is the average GMFI ± SD (*n* = 4) (**F**). * *p* < 0.05, ** *p* < 0.01, *** *p* < 0.005 and **** *p* < 0.0001, ns: no significance.

**Figure 5 pharmaceutics-12-01138-f005:**
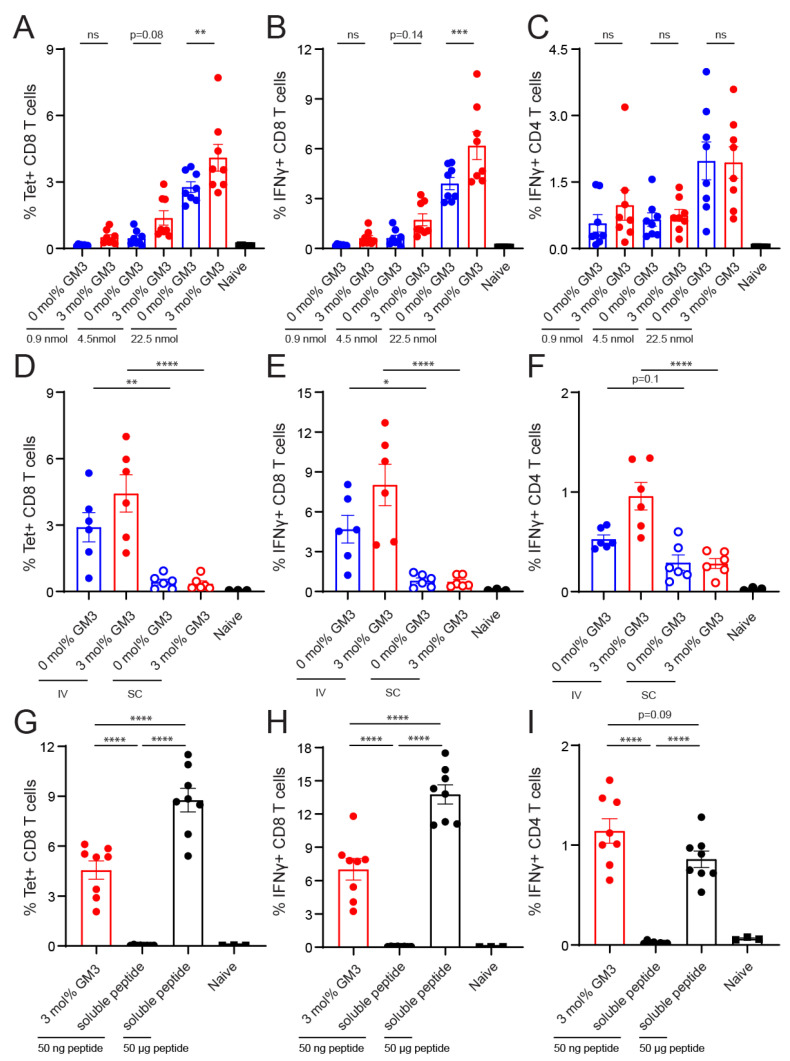
Intravenous administration of GM3-containing liposomes augments vaccine effectiveness. (**A**–**C**) Liposomes were IV administered to mice (22.5, 4.5 and 0.9 nmol of phospholipid for the high, middle and low doses, respectively), supplemented with adjuvant. (**D**–**F**) Liposomes were IV or subcutaneously (SC) administered to mice (22.5 nmol of phospholipid), supplemented with adjuvant. (**G**–**I**) GM3-containing liposomes with peptide (22.5 nmol of phospholipid, which equals 50 ng of peptide) or a low or high dose of free peptide (50 ng and 50 μg, respectively) were IV administered to mice, supplemented with adjuvant. (**A**,**D**,**G**) On Day 7, H-2K^b^/SIINFEKL tetramer binding T cells were identified. (**B**,**C**,**E**,**F**,**H**,**I**) Splenocytes were restimulated with peptide for 5 h or 25 h, for CD8^+^ or CD4^+^ T-cell responses, respectively. Subsequently, IFNγ-producing CD8^+^ (**B**,**E**,**H**) or CD4^+^ (**C**,**F**,**I**) T cells were detected with intracellular flow cytometry staining. Indicated is the GMFI ± SEM (*n* = 8) (pooled from 2 independent experiments) for A–C, (*n* = 6) for the experimental and (*n* = 3) for the naïve group for D–F and (*n* = 8) for the experimental and (*n* = 3) for the naïve group for G–I. * *p* < 0.05, ** *p* < 0.01, *** *p* < 0.005 and **** *p* < 0.0001, ns: no significance.

**Table 1 pharmaceutics-12-01138-t001:** Sequence and molecular mass of OVA-peptides. OVA_247–279_ and fluorescent 5-FAM-Lys-OVA_247–279_ were synthesized in house. Peptide identity was confirmed using mass spectrometry analysis.

Peptide	Sequence	Theoretical Molecular Mass (Da)	Ionization Mode	Observed [M ± 3H]^3±^ (*m*/*z*)	Observed Molecular Mass (Da)
OVA_247–279_	NH2-DEVSGLEQLESIINFEKLT EWTSSNVMEERKIK-COOH	3883.3	Positive	1295.3	3882.9
5-FAM-Lys-OVA_247–279_	NH2-5-FAM-Lys-DEVSGLEQLESIINFEKLTEWTSSNVMEERKIK-COOH	4369.8	Negative	1455.3	4368.9

**Table 2 pharmaceutics-12-01138-t002:** Liposome characteristics of liposomes containing 0–5 mol% of targeting molecule GM3. Liposomal size, polydispersity index (PDI) and zeta potential were measured in 10 or 0.3 mM HEPES saline buffer, pH 7.4, at 25 °C, respectively.

GM3 Amount Incorporated (mol%)	Mean Size (nm)	Mean PDI	Mean Zeta Potential (mV)
0 mol% GM3	169 ± 0.3	0.13 ± 0.02	−61.0 ± 2.7
1 mol% GM3	178 ± 2.2	0.17 ± 0.004	−58.5 ± 1.4
3 mol% GM3	154 ± 0.8	0.17 ± 0.02	−67.3 ± 2.4
5 mol% GM3	158 ± 0.9	0.17 ± 0.006	−64.9 ± 3.0

**Table 3 pharmaceutics-12-01138-t003:** Liposome characteristics of liposomes containing varying mol% of GM3 and/or PEG. Liposomal size, PDI and zeta potential were measured in 10 or 0.3 mM HEPES saline buffer, pH 7.4, at 25 °C, respectively.

GM3/PEG Amount Incorporated (mol%)	Mean Size (nm)	Mean PDI	Mean Zeta Potential (mV)
0 mol% GM3/0 mol% PEG	191 ± 3.9	0.1 ± 0.02	−50.4 ± 2.3
3 mol% GM3/0 mol% PEG	176 ± 1.5	0.1 ± 0.001	−53.2 ± 4.9
0 mol% GM3/5 mol% PEG	159 ± 5.9	0.1 ± 0.05	−24.6 ± 0.7
3 mol% GM3/5 mol% PEG	164 ± 3.7	0.1 ± 0.02	−26.9 ± 1.3

**Table 4 pharmaceutics-12-01138-t004:** Liposome characteristics of small and large control and GM3-containing liposomes. Liposomal size, PDI and zeta potential were measured in 10 or 0.3 mM HEPES saline buffer, pH 7.4, at 25 °C, respectively.

Liposome Size and GM3 Amount Incorporated (mol%)	Mean Size (nm)	Mean PDI	Mean Zeta Potential (mV)
Small 0 mol% GM3	180 ± 0.9	0.1 ± 0.04	−49.6 ± 1.0
Small 3 mol% GM3	167 ± 4.1	0.1 ± 0.02	−52.6 ± 2.6
Large 0 mol% GM3	385 ± 12.4	0.2 ± 0.03	−44.8 ± 1.0
Large 3 mol% GM3	390 ± 12.7	0.2 ± 0.06	−46.2 ± 0.8

**Table 5 pharmaceutics-12-01138-t005:** Liposome characteristics of OVA-peptide-containing liposomes. Liposomal size, PDI and zeta potential were measured in 10 or 1 mM HEPES saline buffer, pH 7.4, at 25 °C, respectively. Peptide loading efficiency and capacity were analyzed using fluorescent 5-FAM-Lys-OVA_247–279_.

GM3 Amount Incorporated (mol%)	Mean Size (nm)	Mean PDI	Mean Zeta-Potential (mV)	Loading Efficiency (%)	Loading Capacity (ng Peptide/nmol Phospholipid)
0 mol% GM3	231 ± 6.1	0.2 ± 0.05	−64.8 ± 2.3	0.40 ± 0.01	2.40 ± 0.04
3 mol% GM3	224 ± 15.5	0.2 ± 0.05	−44.5 ± 1.8	0.43 ± 0.02	2.41 ± 0.09

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
