# Peer review of "Optimization of Liposomes for Antigen Targeting to Splenic CD169+ Macrophages"

_pharmaceutics, 2020, doi:10.3390/pharmaceutics12121138_

Round 1
Reviewer 1 Report
The study aimed to assess how the density of GM3 on the surface of liposomes, the presence of PEG and liposomal size affect binding and uptake into CD169+ macrophages in vitro and in vivo. The work appears novel, and with the ever-increasing interest in the development of cancer vaccines and immunotherapies, timely, and of broad interest to the readers of the journal Pharmaceutics.
The authors should address the minor comments below before the manuscript is considered suitable for publication.
General comments/questions
- Splenic and lymph node CD169+ MΦ are known for their capacity to capture particles, why were the lymph nodes not examined for uptake/ CD8+ and CD4 + T cell populations/response?
- Why were functionalised (3 mol% and 5 mol% GM3) liposomes smaller than non-funtionalised liposomes?
- Could CD169+ MΦ uptake be related to liposome charge?
- With regards to your Eliza results, how did you confirm that excess GM3 (GM3 that was not incorporated into the liposome) was successfully removed?
- It is becoming increasingly important to accurately characterise functionalised liposome formulations in order to translate the findings to the clinic. Please touch on methods you could use to quantify GM3 density on the surface of liposomes in future work, see the following refs:
- Belfiore, L., L. M. Spenkelink, M. Ranson, A. M. van Oijen, and K. L. Vine. 2018. "Quantification of ligand density and stoichiometry on the surface of liposomes using single-molecule fluorescence imaging." J Control Release 278:80-86. doi: 10.1016/j.jconrel.2018.03.022.
- Dai, Qin, Stefan Wilhelm, Ding Ding, Abdullah Muhammad Syed, Shrey Sindhwani, Yuwei Zhang, Yih Yang Chen, Presley MacMillan, and Warren C. W. Chan. 2018. "Quantifying the Ligand-Coated Nanoparticle Delivery to Cancer Cells in Solid Tumors." ACS Nano 12 (8):8423-8435. doi: 10.1021/acsnano.8b03900.
Specific comments/corrections
Page 3, line 119 ‘ treated with 2% Tween X-100’ please clarify if liposomes were treated with triton X-100 or Tween or a combination of both?
Page 4, line 137 Please include cell lines details for THP-1, from what cancer were they originally derived, where were they obtained (ie gifted or ATCC etc), were they screened and deemed negative for mycoplasma, what passage number were the cells when experiments were preformed?
Page 4, line 147 please include the details of the strains(s) of mice spleens were harvested from or move the ‘animal experiments’ section above.
Page 5, Animal experiment section- please add approved ethics/protocol number and the institutional ethics committee that approved their use.
Author Response
Please find our reply attached.

Reviewer 2 Report
Twilhaar and co-authors described the formulation of liposomes for the antigen targeting of splenic CD169+ macrophages using GM3 ganglioside. They showed how the size and the PEG coating affected the ability of GM3-coated or not coated liposomes to be internalized in splenic CD169+ macrophages. Moreover they showed how the optimized formulation was able to induce vaccination in vivo with liposomes loaded with OVA peptides. The work in general is well conducted and well presented and I suggested the publication in Pharmaceutics after minor revisions:
-Please check the entire text for grammar mistakes and typos.
-A brief description of OVA peptides should be added in the Introduction paragraph.
-In the entire text there are some acronyms that should be clarify at the first mention.
-Should be interesting understand if lower mol% of PEG could increase lifespan of liposomes in vivo without affecting GM3 targeting capabilities.
-Is not clear in the Results section why the authors decided to produce liposomes with a double diameter respect to the original one. They should clarify this point also here and not only in the discussion. Why they also did not try to decrease liposomes size?
Author Response
Please find our reply attached.

Reviewer 3 Report
Manuscript ID: pharmaceutics-1001766
The manuscript of Maarten Nijen Twilhaar, Lucas Czentner, Joanna Grabowska, Alsya Affandi, Jerry Lau, Katarina Olesek, Hakan Kalay, Cornelus van Nostrum, Yvette van Kooyk, Gert Storm and Joke M.M. den Haan as Co-authors: “Optimization of liposomes for antigen targeting to splenic CD169+ macrophages” presents the studies regarding incorporation of physiological ligand for CD169, the ganglioside GM3, into liposomes to enhance liposome uptake by CD169+ macrophages.
The subject is of interest and would be useful for researchers working in the fields. As I am not a strong expert in pharmacology my suggestions mainly concern the production and characterization of liposomes. However, there are some issues needed to be added and addressed before publication.
Major issues:
- Part of the authors of this manuscript are also the authors of another thematically closely related manuscript: Selective tumor antigen vaccine delivery to human CD169+ antigen-presenting cells using ganglioside-liposomes. Alsya J. Affandi, Joanna Grabowska, Katarzyna Olesek, Miguel Lopez Venegas, Arnaud Barbaria, Ernesto Rodríguez, Patrick P. G. Mulder, Helen J. Pijffers, Martino Ambrosini, Hakan Kalay, Tom O’Toole, Eline S. Zwart, Geert Kazemier, Kamran Nazmi, Floris J. Bikker, Johannes Stöckl, Alfons J. M. van den Eertwegh, Tanja D. de Gruijl, Gert Storm, Yvette van Kooyk, Joke M. M. den Haan. Proceedings of the National Academy of Sciences.t 2020, 202006186; DOI: 10.1073/pnas.2006186117. However, the authors did not mention this work and did not compare and analyse the obtained results with the previously published. This point has to be justified in the manuscript.
- As the title of the manuscript is “Optimization of liposomes..., please remember it and put stress through all manuscript on the procedure of optimization of liposomes. In this edition of the manuscript preparation of liposomes and their characterisation is described quite poorly, in the manner - so, by the way....
- Paragraph Materials and Methods, subparagraph Liposome preparation. Please give a more detailed description of a methodology which was used for liposome preparation and phosphate content determination.
- Please justify, characterisation of liposomes was performed for freshly prepared samples or after some storage. In the case of storage please provide conditions for the
- Please provide also the detailed description of the conditions of DLS experiment, add missing data - Software, specifications: medium, refractive index, viscosity, dielectric constant, nanoparticles, the refractive index of materials, wavelength. How many times the measurements were performed?
- Please justify the choice and ratio of used lipids.
- Please justify used amount of GM3. Why authors used 0-5% with step 0, 1, 3 and 5 mol%?
- In the lines 247-249 authors concluded that “...these results indicate that in vitro 3-5 mol% GM3-containing liposomes exhibit superior binding properties to both mouse and human CD169-expressing cells, compared to control liposomes that lack GM3.” However for further studies of PEG influence on liposomes only 0 and 3 mol% GM3-containing liposomes were used. Why the authors have chosen only the liposomes with 3 mol% GM3 for further studies? What about liposomes with 5 mol% GM3?
- Please justify chosen amount of PEG.
- Please underline the difference (preparation, storage, what else???) for the samples defined as 0 mol% GM3 and 3 mol% GM3 from Table 2 and samples Small 0 mol% GM3 Small 3 mol% GM3, Large 0 mol% GM3, Large 3 mol% GM3 from Table 4?
- It seems that ones from the sample pairs - 0 mol% GM3 and Small 0 mol% GM3, as well as 3 mol% GM3 and Small 3 mol% GM3 are more or less the same. Please give your explanation regarding the different mean size and zeta potential values between both samples in the corresponding pairs.
- In conclusion please provide some recommendations and justify the necessity of the components of the liposome or any other relationships regarding optimisation of liposomes.
Minor issues:
- Please check style and use the same style for the description of the degree of Celsius.
- Line 139, CO2, please use subscript for 2.
- Please write values of zeta potential in all tables in the same style using the same of significant digits.
Based on the considerations detailed above, I do recommend accepting this manuscript for publication with major revision.
Author Response
Please find our reply attached.

Round 2
Reviewer 3 Report
accept for publication